# Analysis of Continual Learning Techniques for Image Generative Models with Learned Class Information Management

**DOI:** 10.3390/s24103087

**Published:** 2024-05-13

**Authors:** Taro Togo, Ren Togo, Keisuke Maeda, Takahiro Ogawa, Miki Haseyama

**Affiliations:** 1Graduate School of Information Science and Technology, Hokkaido University, N-14, W-9, Kita-ku, Sapporo 060-0814, Hokkaido, Japan; taro_togo@lmd.ist.hokudai.ac.jp; 2Faculty of Information Science and Technology, Hokkaido University, N-14, W-9, Kita-ku, Sapporo 060-0814, Hokkaido, Japan; togo@lmd.ist.hokudai.ac.jp (R.T.); ogawa@lmd.ist.hokudai.ac.jp (T.O.); 3Data-Driven Interdisciplinary Research Emergence Department, Hokkaido University, N-14, W-9, Kita-ku, Sapporo 060-0814, Hokkaido, Japan; maeda@lmd.ist.hokudai.ac.jp

**Keywords:** continual learning, selective amnesia, machine unlearning, generative model

## Abstract

The advancements in deep learning have significantly enhanced the capability of image generation models to produce images aligned with human intentions. However, training and adapting these models to new data and tasks remain challenging because of their complexity and the risk of catastrophic forgetting. This study proposes a method for addressing these challenges involving the application of class-replacement techniques within a continual learning framework. This method utilizes selective amnesia (SA) to efficiently replace existing classes with new ones while retaining crucial information. This approach improves the model’s adaptability to evolving data environments while preventing the loss of past information. We conducted a detailed evaluation of class-replacement techniques, examining their impact on the “class incremental learning” performance of models and exploring their applicability in various scenarios. The experimental results demonstrated that our proposed method could enhance the learning efficiency and long-term performance of image generation models. This study broadens the application scope of image generation technology and supports the continual improvement and adaptability of corresponding models.

## 1. Introduction

The rapid advances in deep learning technology have enabled the development of various models for different applications [1,2,3]. These models often equal or exceed human capabilities, particularly in the fields of vision and language. Specifically, image generation models are gaining attention for their ability to generate desired images from text prompts [4]. A wide variety of models such as Midjourney (https://www.midjourney.com/home (accessed on 20 March 2024)) are available to the public and can be employed by users to generate images. However, training these models requires an enormous amount of time and effort [5], and the process of adapting these models to new data and tasks is often quite costly and inefficient [6].

Continual learning provides an effective solution to these challenges. The goal of this approach is to extend the lifespan of a trained model and maintain its effectiveness over time by improving the model’s adaptability to new tasks. Notably, with this approach, when a model adapts to new tasks, it retains existing knowledge and modifies already gained knowledge without compromising performance. Continual learning enables the model to maintain and update its knowledge over time consistently, ensuring its effectiveness and suitability for tasks. Class incremental learning (CIL) is a process through which a model adapts to gradually increasing new classes during continual learning [7]. A major hindrance to CIL is catastrophic forgetting [8], which is a phenomenon where the model forgets existing classes while learning new classes. Catastrophic forgetting degrades the performance as the model loses knowledge of previous classes [6]. Therefore, to achieve CIL, approaches that allow the model to process new knowledge while preventing this phenomenon are vital. To address this issue, several methods [9,10] have been proposed, including the introduction of memory systems that preserve the key features of existing information and the development of algorithms that integrate knowledge between new and old classes in a balanced way. Thus, CIL aims to enhance the ability of models to adapt to new knowledge while retaining existing knowledge for continual learning capabilities. This approach is becoming increasingly important for developing machine learning models that respond to dynamically changing data and new tasks.

Meanwhile, machine unlearning is the intentional removal of specific information from an existing model during continual learning. Furthermore, this process becomes more important when a model contains harmful information or data protected by copyright. The removal of unnecessary information from a trained model often involves removing data from the original dataset and retraining the model from scratch, which are time-consuming and cost-intensive. Therefore, several approaches have been developed to efficiently remove specific information without retraining the model from scratch [11,12,13,14,15]. These methods segment certain parts of the model or change the learning process to reduce costs while maintaining the model accuracy. Additionally, they not only enable the model to stay updated and meet social responsibilities and legal requirements but also allow for its continual use over a long period.

Selective amnesia (SA) [16] has been proposed as a machine learning method. It performs the effective replacement of specific information with new information in image generation models. This technique is designed to replace specific information while the model retains necessary information by combining two continual learning strategies: generative replay (GR) [15] and elastic weight consolidation (EWC) [17]. GR regenerates data learned from previous tasks and provides the necessary information. EWC, in contrast, adjusts weight updates to protect existing information when the model replaces unnecessary information with new knowledge. This combination allows SA to replace specific information and successfully minimizes the effect of replacement on other information. This process is similar to the brain’s forgetting mechanism and cognitive processes [18], ignoring unnecessary information and retaining only necessary information. This process also highlights the importance of models replacing and forgetting existing information while incorporating new knowledge appropriately.

In a CIL context, replacement methods are very useful. When new classes are continually added during CIL, it is assumed that the existing class information is relevant for learning classes. Continually replacing unnecessary classes with new ones could potentially improve a model’s performance and flexibility in CIL. Thus, care must be taken in deciding which classes to replace and in assessing the effect of replacement on other classes. In particular, the outstanding generative capability of generative models has led to an increase in the number of users, and society’s demand for the removal of harmful images and output to suit individual preferences has increased correspondingly. However, such models are so large that it is difficult to train them from scratch. Therefore, further research to address these problems using replacement techniques is needed. This approach could extend the application range of continual learning and support the continual evolution and adaptability of generative models.

Here, we attempt to improve the continual use of image generation models by introducing class replacement and forgetting techniques. As shown in Figure 1, we replace unnecessary classes with new classes. In another approach, we replace unnecessary classes with the class of noise images and facilitate the learning of new classes. This study uses modified SA. In previous research, this method has been proposed as a machine unlearning method for some of a model’s knowledge. However, in this study, the existing classes are replaced by noise images or new classes, and SA is used to facilitate the learning of new classes. We apply the proposed method and analyze the performance of the image generation models in learning new classes. Additionally, we examine the impact of forgetting by replacing unnecessary classes with the class of noise images as well as the impact of learning new classes. Our proposed method uses class-replacement techniques to update class information and unnecessary classes in pretrained image generation models. This allows the model to learn and retain new classes more efficiently in evolving data environments, thereby enhancing their adaptability and long-term performance in image generation fields.

The main contributions of this study are summarized in three points.

**Detailed analysis of the class-replacement technique:** We introduce class-replacement techniques for image generation models and evaluate their performances experimentally using multiple datasets.

**Discussion of class-replacement techniques:** After proving experimentally that the use of unnecessary classes improves the CIL performance, we discuss the functioning of the model in relation to our method.

**Exploring class-replacement techniques:** We explore the effectiveness and potential of class-replacement techniques in various scenarios to expand the application scope of image generation models. We discuss the applicability and flexibility of our proposed method under the realistic use of image generation models and confirm its effectiveness.

This research will promote the development of image generation models to expand their application scope to various data-driven fields. Our proposed method is expected to enable the long-term use and continual performance improvement of models, opening new horizons in image generation.

## 2. Related Work

This section provides an overview of existing research in the fields of continual learning, CIL, generative models, and machine unlearning. We also explain how these areas are interrelated and specify their relevance in developing approaches to address the challenges of continual learning in image generation models.

### 2.1. Continual Learning

Continual learning refers to the process by which a model continuously learns and adapts to new tasks and data over time. It encompasses the removal of unnecessary information or management of information to maintain freshness and robustness. This process is especially crucial in dynamic environments where data types change and new requirements for the model arise [6]. The objective of continual learning is to make models robust and adaptive to the constantly changing conditions of real-world scenarios, which cannot be archived using static training methods.

In general, continual learning assumes a scenario where new data are generated and the learned knowledge is retained to handle the new task data. In a broad context, continual learning involves not only adaptation to new data but also the strategic removal of outdated or unnecessary information. This approach is essential for keeping models up to date and efficient in today’s dynamically changing society, where new demands for models emerge constantly. Consequently, various strategies have been employed to address these challenges, including model manipulation based on data cashing and model reorganization [19]. These strategies enhance the ability of models to remain productive and ensure efficient resource utilization without training the models from scratch. In this context, SA successfully eliminates unnecessary information by replacing existing information in the model with different knowledge [16]. While various methods have been proposed for eliminating existing information from a model, SA is superior to them as it does not require any access to the original data or learning processes.

### 2.2. Class Incremental Learning

It is important to learn new knowledge that the model does not have. The ability to learn new knowledge keeps the model up to date and allows it to meet more demands as it grows. However, when models learn new knowledge [20], there is the problem of catastrophic forgetting, whereby they lose previously learned knowledge as they learn new knowledge. Moreover, the field faces a stability–plasticity dilemma associated with striking a balance between the absorption of new knowledge and the retention of existing information.

As a form of continual learning, CIL handles situations where new classes are introduced over time [6]. This approach aims to update the model to learn the newly added classes accurately while retaining information about the existing classes. Several methods [15,21] have been successful in ensuring that the model does not forget previous class information when learning new knowledge. Such methods often involve allowing the model to regenerate the training data and restrict the loss function. Conversely, no study has attempted to validate a model that addresses both the incorporation of new information and the erasure of outdated data, highlighting the need for innovative solutions.

### 2.3. Generative Models

The evolution of deep-learning-based image generation models has been particularly remarkable in the field of image generation. In this field, various algorithms that can produce more detailed images than traditional models have been developed. This paper discusses the characteristics, challenges, and prospects of the following mainstream image generation models: variational autoencoders (VAEs) [22], generative adversarial networks (GANs) [23], and diffusion models [24].

#### 2.3.1. VAE: Image Generation Based on a Latent Variable Space

VAEs [22] are a type of generative model that map images to a latent variable space, from which new images can be synthesized. This latent space serves to represent the intrinsic features of data in a lower-dimensional form and is rooted in the concept of dimensionality reduction in machine learning. The research on VAEs has made significant advancements. For instance, conditional VAEs [25,26] can generate images with specific characteristics based on given conditions, enabling the generation of the intended images. β-VAE [27] focuses on enhancing the interpretability of the latent space, clarifying which aspects of an image correspond to which parts of the latent variable space, thus enabling more controlled image generation. These advancements not only improve image quality but also illustrate the importance of learning interpretable representations within generative models [28]. The applications of VAEs are expected to extend beyond mere image generation to include various types of data type conversion, anomaly detection, and data imputation [29,30,31]. Through exploring, understanding, and manipulating latent spaces, we can determine the contribution of these applications to solve real-world problems, and we will outline the next stages of research in this area [32].

#### 2.3.2. GAN: High-Resolution Image Generation through Adversarial Learning

GANs [23] comprise a pair of neural networks, i.e., a generator and a discriminator, each serving distinct functions. The generator is trained to emulate the data distribution learned from actual images to fabricate new images. The generator and discriminator engage in a continual feedback loop of competition, where the generator improves its ability to produce realistic images and the discriminator becomes more proficient at detecting sophisticated forgeries. This optimization process, driven by the discriminator’s evaluation of the generator, is termed adversarial learning [33]. Research on GANs has spanned various dimensions, leading to numerous enhancements. For instance, conditional GANs (cGANs) can generate images based on specific labels or attributes, enabling more controlled image creation. Meanwhile, cycleGANs [34,35] facilitate style translation between different domains, allowing for unique applications, such as the transformation of horses into zebras. These advancements underscore the versatility and efficacy of GANs for particular applications [36]. However, GANs encounter substantial challenges, such as mode collapse and training instability; research is in progress to find solutions. Future research on GANs is expected to address these issues while emphasizing the quality of the images generated. For example, a progressive GAN [37] has been successfully employed to produce high-resolution images by incrementally increasing the image sharpness. Techniques such as BigGAN [38] utilize more extensive networks and training data to produce remarkably realistic images.

#### 2.3.3. Diffusion Models: Pioneering New Possibilities in Image Generation

In recent years, diffusion models have gained remarkable attention as a novel class of generative models. These models initiate with random noise and progressively form images by adhering to the data distribution, thereby facilitating the generation of high-quality images. This process comprises two phases: a diffusion process, where noise is gradually added to images, and a reverse diffusion process, where noise is incrementally removed to restore the original image. One of the advantages of this approach is its simplicity and ease of training, as it does not require a latent variable space. The application scope of diffusion models is broad, encompassing not only image generation but also such tasks as image restoration and super-resolution. A denoising diffusion probabilistic model [39] is one example of a diffusion model that has demonstrated exceptional performance in image generation, capturing complex data distributions effectively. Furthermore, improved denoising diffusion probabilistic models [40], which exhibit an enhanced generation efficiency and image quality, have been proposed. One of the main challenges associated with diffusion models is their time-consuming image generation process; this is because the reverse diffusion process requires numerous steps. This can be problematic for real-time applications. To address this issue, research has focused on developing acceleration techniques and efficient algorithms [41]. In addition, future research is expected to focus not only on the efficiency of diffusion models but also on expanding their application scope and improving the quality of generated images [41,42].

#### 2.3.4. Future of Image Generation Models

Image generation models hold significant potential for application in various fields, including entertainment, healthcare, and education. These models can provide novel visual experiences by generating realistic images, can assist in medical image analysis, and can serve as valuable tools in educational settings for creating visual materials [43]. In the entertainment industry, they can create realistic backgrounds and characters for movies and games. The applications of generative models are expected to extend beyond image generation to fields such as video production and 3D modeling [44,45]. Particularly, the video-generation artificial intelligence model *Sora* (https://openai.com/sora (accessed on 20 March 2024)), presented by OpenAI, has shown considerable potential. Continuous research and technological innovation will facilitate further advancements in image generation technologies, enriching our lives in various ways.

### 2.4. Machine Unlearning

Machine unlearning is the process of effectively removing specific data or information from a model [11,12]. This concept is particularly important for privacy protection, data deletion requests (e.g., “right to be forgotten” (https://gdpr.eu/right-to-be-forgotten/ (accessed on 20 March 2024))), or simple outdated or incorrect information removal from the model. Unlike CIL, machine unlearning focuses on forgetting information that has already been learned. Machine unlearning, such as model splitting during the training process, can enable models to effectively forget information while maintaining existing models [46,47]. In the model-splitting approach, one large model is split into many smaller submodels or modules. To remove specific data, only the relevant submodels are retrained or adjusted. The main advantage of this approach is that it reduces computational resource consumption without having to retrain the entire model from scratch.

These approaches allow models to continually evolve and adapt to new datasets and tasks while effectively forgetting outdated or unnecessary information. Consequently, machine unlearning plays an important role in the long-term use of models. It has various uses, including privacy protection and inaccurate information removal.

#### Selective Amnesia (SA): Forgetting Approach

SA, a technique to induce forgetfulness in image generation models, has gained attention in recent years. This method aims to prevent generative models from producing harmful content. Traditional approaches have considered excluding specific concepts from training datasets to avoid generating inappropriate content. However, this filtering is practically challenging, and trained models should be able to forget certain concepts. SA approaches forgetfulness from the perspective of continual learning. In continual learning, the goal is to prevent models from forgetting previously learned tasks when learning a new task. SA aims to make a trained model forget only the concepts of a specific task. While various machine unlearning methods have been studied, as described in previous chapters, SA is superior to them as it does not require access to the original dataset and has access to the original learning process and architecture.

This study takes a different approach from SA and focuses on the effect of correcting the discrimination space of trained models in classification problems on continual learning. The proposed method applies the forgetfulness mechanism as a requirement when learning new tasks, thereby enabling the efficient learning of new tasks, even for models with limited capacity. Additionally, the method aims to facilitate the learning of new tasks and prevent forgetting by appropriately correcting the discrimination space of the trained model.

### 2.5. Remaining Problems and Our Approach

Although much progress has been made with continual learning and machine unlearning, there is a lack of research on how machine unlearning affects the learning of new knowledge. It is important to understand this interaction, as forgetting knowledge may indirectly influence the learning and retention of new information. SA has been proposed as a general machine unlearning method, focusing mainly on forgetting. However, SA is also useful in continual learning as it enables forgetting without changing the architecture of the model or accessing the learning process. This research aims to integrate continual learning and machine unlearning and investigate the impact of forgetting on the learning of new knowledge.

## 3. Proposed Methods

We propose two methods for modifying deep generative models in CIL: *replace* and *forget and learn*, as shown in Figure 2. The *replace* method involves the direct replacement of an unnecessary class with a new class, whereas the *forget and learn* method involves replacing the unnecessary class with a noise image, followed by the learning of a new class using a continual learning approach. We consider a generative model *M* capable of producing images of classes $c_{1},c_{2},\ldots ,c_{n}\in \begin{array}{c} C_{class} \end{array}$ (Symbols and notations are explained in Appendix A). This model is designed to sequentially replace or forget unnecessary classes. Let $c_{f}$ be the class that becomes invalid or unnecessary. In the *replace* method, by replacing $c_{f}$ with $c_{\mathrm{new}}$, we can erase $c_{f}$ and obtain a final model, $M_{\mathrm{final}}$. Conversely, in the *forget and learn* method, the unnecessary class, $c_{f}$, is replaced with the class of noise image $c_{\mathrm{noise}}$, after which training is adapted for the new class, $c_{\mathrm{new}}$. The final model, $M_{\mathrm{final}}$, is trained to be able to generate images of the newly added class, $c_{\mathrm{new}}$, and the retained classes, $c_{r}$. In this method, the dataset is described as follows:(1)$$\begin{array}{cc} D & =D_{f}\cup D_{r}\cup D_{\mathrm{new}}\cup D_{\mathrm{noise}}\\ \; & ={\left\{\left(x_{f}^{\left(n\right)},c_{f}^{\left(n\right)}\right)\right\}}_{n}\cup {\left\{\left(x_{r}^{\left(m\right)},c_{r}^{\left(m\right)}\right)\right\}}_{m}\cup {\left\{\left(x_{\mathrm{new}}^{\left(l\right)},c_{\mathrm{new}}^{\left(l\right)}\right)\right\}}_{l}\cup {\left\{\left(x_{\mathrm{noise}}^{\left(o\right)},c_{\mathrm{noise}}^{\left(o\right)}\right)\right\}}_{o}, \end{array}$$
where $D_{f}$, $D_{r}$, $D_{\mathrm{new}}$, and $D_{\mathrm{noise}}$, respectively, represent the datasets of forgotten class images, retained class images, new class images, and noise class images, and ${\left\{\left(x^{\left(i\right)},c^{\left(i\right)}\right)\right\}}_{i}$ denotes the pair of the *i*th image, *x*, and its corresponding class, *c*. $D_{f}$ and $D_{r}$ are defined optionally by the user, $D_{f}$ usually contains copyrighted or harmful data, and $D_{\mathrm{new}}$ is selected as newly required data for the new task. The overall distribution of *D* is a joint distribution denoted by p(x,c)=p(x|c)p(c). Further, the distribution across class labels is defined as $p\left(c\right)=\sum_{i\in f,r,\mathrm{new},\mathrm{noise}}p_{i}\left(c_{i}\right)$.

The proposed method uses a modified SA to develop a replaced and forgotten model. SA allows the replacement of existing classes with specific knowledge or meaningless image information, such as noise images in pretrained models. Our approach experimentally examines the use of such replacements and forgetting approaches in CIL to improve the performance of image generation models.

We assume a pretrained image generation model parameterized by $\theta^{*}$ = arg $\max_{\theta }$ [ $E_{p(x_{c})}\log p\left(x|\theta ,c\right)$], which represents the maximum likelihood estimate for the dataset, *D*. For simplicity, we use subscripts for distributions and class labels, such as $p_{f}\left(c\right)$ and $p(c_{f})$, where θ represents the model. The goal of the *replace* method is to adapt this model to the generation of $D_{\mathrm{new}}|c_{f}$ while maintaining the generation of $D_{r}|c_{r}$, whereas the *forget and learn* method aims to adapt this model to the generation of $D_{\mathrm{new}}|c_{\mathrm{new}}$ while maintaining the generation of $D_{r}|c_{r}$.

From a Bayesian viewpoint, the *replace* method is inspired by the formulation of EWC and described as
(2)$$\begin{array}{cc} \log p(\theta |D_{\mathrm{new}},D_{r}) & =\log p\left(\theta |D_{\mathrm{new}}\right)+\log p\left(\theta |D_{r}\right)-\log p\left(\theta |D_{\mathrm{new}},D_{r}\right)\\ \; & =\log p\left(D_{\mathrm{new}}|\theta \right)+\log p\left(\theta |D_{r}\right)+\begin{array}{c} Const \end{array}. \end{array}$$
Here, Const is a constant that arises from the normalization of probabilities. For retaining the class information, we are interested in the posterior conditioned only on $D_{r}$, (3)$$\begin{array}{cc} \log p(\theta |D_{r}) & =-\log p\left(D_{\mathrm{new}}|\theta \right)+\log p\left(\theta |D_{\mathrm{new}},D_{r}\right)+\begin{array}{c} Const \end{array}\\ \; & =-\log p\left(x_{\mathrm{new}}|\theta ,c_{f}\right)-\lambda \sum_{i}\frac{F_{i}}{2}{\left(\theta_{i}-\theta_{i}^{*}\right)}^{2}+\begin{array}{c} Const \end{array}, \end{array}$$
where $\log p\left(D_{\mathrm{new}}|\theta \right)=\log p\left(x_{\mathrm{new}}|\theta ,c_{f}\right)=\log p\left(x_{\mathrm{new}}|c_{f}\right)+\log p\left(c_{f}\right)$ so that the conditional likelihood is unnecessary. We substitute $\log p(\theta |D_{r},D_{\mathrm{new}})$ with the Laplace approximation of EWC; $F_{i}$ is the Fisher information matrix (FIM). λ is a weight parameter and controls the balance of $D_{r}$ and its adaptation to $D_{new}$. As λ increases, $D_{r}$ remains clear but lacks diversity and is less adaptable to $D_{new}$ [16]. To retain $c_{r}$, we maximize $\log p(\theta |D_{r})$ to obtain a maximum posterior estimate. Intuitively, maximizing Equation (Equation 2) lowers the likelihood of $\log p(x_{\mathrm{new}}|c_{f})$ while keeping θ close to $\theta^{*}$.

The optimization objective in Equation (Equation 2) does not need to use samples from $D_{r}$. Without the original data from $D_{r}$, the model’s ability to generate the data to be remembered diminishes over time. Although an evidence lower bound (ELBO) is available, minimizing this lower bound does not necessarily decrease the log-likelihood. Therefore, we focus on variational models where the log-likelihood is intractable. Both efforts are achieved using generative replay and a surrogate objective (ELBO), as proposed in [48,49]. Thus, the *replace* method can replace $c_{f}$ with $c_{\mathrm{new}}$.

In the *forget* approach of the *forget and learn* method, $D_{\mathrm{new}}$ and $x_{\mathrm{new}}$ in Equation (Equation 2) are $D_{\mathrm{noise}}$ and $x_{\mathrm{noise}}$, respectively. In this approach, the model replaces existing class information with noise images, facilitating the process of forgetting old information. In the next *learn* approach, the model learns $c_{\mathrm{new}}$ using a CIL method such as EWC and GR. In practice, the two methods, *replace* and *forget and learn*, differ significantly, as shown in Figure 2. The *replace* method and the *forget and learn* method differ in the placement of $c_{\mathrm{new}}$ in the latent space. Specifically, the *replace* method replaces $c_{f}$ with $c_{\mathrm{new}}$ in the latent space, while the *forget and learn* method learns $c_{\mathrm{new}}$ between existing classes. When the datasets are incorrect or information needs to be updated, the *replace* method updates the model by changing the existing class. Given a particular input $c_{i}$, the desired output for this input is expressed as ${\hat{c}}_{i}$. First, the replacement approach changes the behavior of the model by simply replacing one class with another. Specifically, for a given input class, $c_{f}$, the model replaces it with a previously outputted class ${\hat{c}}_{f}$. After the replacement, the model outputs class ${\hat{c}}_{\mathrm{new}}$ for input class $c_{f}$. This mismatch between the input and output classes requires a translator to deliberately treat input class $c_{f}$ as a different input class, $c_{\mathrm{new}}$, for output class ${\hat{c}}_{\mathrm{new}}$ in practical applications. However, distinguishing the effects in the latent space and translator is challenging, and when updating existing classes, a translator is not necessary. Therefore, a translator was not considered in this study. In contrast, when the information on the existing classes is completely erased and the model for the new input is changed, the *forget and learn* method is needed. The approach to forgetting unnecessary classes is to initially replace the existing classes with the class of noise images and subsequently facilitate the learning of new classes using CIL methods. In this approach, the model outputs noise images for input class $c_{f}$, whereas it correctly outputs ${\hat{c}}_{\mathrm{new}}$ for the new input class, $c_{\mathrm{new}}$.

## 4. Experiments

To verify the effectiveness of the proposed method, we applied it to several datasets. By observing the impact of replacing or forgetting classes at each checkpoint on other classes, the effectiveness of the proposed method was quantitatively and qualitatively assessed.

### 4.1. Datasets

To evaluate and analyze the model’s performance, the MNIST (http://yann.lecun.com/exdb/mnist/ (accessed on 20 March 2024)) and Fashion-MNIST (https://github.com/zalandoresearch/fashion-mnist?tab=readme-ov-fil (accessed on 20 March 2024)) datasets were used. These datasets are suitable for analyzing the proposed method in terms of the number of classes and image complexity. These datasets are well known for their class similarities and data structure. We show the details of these two datasets.

MNIST is a dataset of images of handwritten numbers. The dataset contains 10 different numbers from zero to nine and comprises 70,000 greyscale images in total. Each image has a pixel resolution of 28×28, with a number in the center. MNIST is widely used as a basic benchmark for image-recognition techniques.

Fashion-MNIST is a dataset of clothing images and comprises 70,000 greyscale images. Similar to MNIST, the image size is 28×28 pixels. The dataset contains 10 different garments (T-Shirts/Top, Trouser, Pullover, Dress, Coat, Sandal, Shirt, Sneaker, Bag, and Ankle Boot). Fashion-MNIST was designed as a more modern and diverse pattern dataset than MNIST.

### 4.2. Settings

The influence of the proposed method on other classes was verified by calculating its classification accuracy for other classes when replacing and forgetting and when learning new classes on multiple datasets. In addition, the classification accuracy for learning new classes after forgetting each of the 10 classes was calculated to evaluate the effectiveness of the method. Through this process, we verified the robustness of the proposed method for various class characteristics.

We employed a VAE with one-hot encoding as a generative model. This VAE has a simple architecture with two hidden layers. These layers have dimensions of 256 and 512, respectively, and the latent space dimension is set to 8. The output distribution is the Bernoulli distribution, and the prior distribution is the standard Gaussian distribution. This structure allows for a comprehensive and easily accessible evaluation of the effectiveness of the proposed method. The learning process is organized to retain information for nine out of ten classes and thereafter newly learn the remaining one class; learning nine classes requires 100,000 steps. The learning rate is $10^{-4}$, and the batch size is 256. Through this architecture, we verified the robustness of the method for different datasets and class characteristics.

The modified SA technique is employed to perform the replacing and forgetting processes. During replacing, only images of $c_{\mathrm{new}}$ are used as training data, whereas noise images are used as a certain class during the forgetting process. In this learning process, 50,000 samples from the frozen copy of the VAE, from which the nine classes of information are obtained in the previous step, are used to calculate the FIM. For the forgetting process, we employed fine-tuning, a baseline method in CIL, to learn the new class. The training dataset includes nine classes, incorporating both $c_{r}$ and $c_{\mathrm{new}}$. Based on the previous study [16], 100,000 training steps were employed. The learning rate was set to $10^{-4}$, and the batch size was 256. The weight parameter λ was set to 100. $D_{\mathrm{noise}}$ consists of images randomly generated from a uniform distribution between 0 and 255, each being 28 × 28 pixels in size. Each pixel is independently generated, and there is no correlation between pixels.

### 4.3. Evaluation

We evaluate the performance of the model in the “replace”, “forget”, and “learn” phases. The performance of the model is evaluated in two ways:**Qualitative evaluation:** By outputting 10 images from each of the 10 classes for a total of 100 images. We evaluate the quality and diversity of the images generated by the model.**Quantitative evaluation:** Outputting 100,000 images for each 10 class, and then an external classifier classifies them. The classifier can correctly classify 97% of the given datasets. The images are classified into classes by the classifier with high confidence. This approach quantitatively evaluates the images produced by the model based on the accuracy of their classification to a particular class.

Finally, the difference in learning performance between the forgotten and new classes in each dataset is measured for all cases. This allows us to understand the relationship between the characteristics of the classes and the performance of the model in learning the new class when it forgets $c_{f}$.

## 5. Results

### 5.1. Replace Method

Figure 3a shows the experimental results based on the setting $\left(c_{f},c_{\mathrm{new}}\right)=\left(8,3\right)$ for the *replace* method. The green box in this figure shows that the output of $c_{f}$ in the existing class is newly outputted as $c_{\mathrm{new}}$. This indicates that $c_{f}$ in the existing model is no longer available and that $c_{\mathrm{new}}$ is now available. Conversely, the yellow box in this figure shows that the output of $c_{\mathrm{new}}$ in the existing model has not been subjected to relevant learning because no training data have been provided. Figure 3b shows the results of an experiment in which a similar setup, i.e., $\left(c_{f},c_{\mathrm{new}}\right)=\left(\mathrm{Bag},\mathrm{Dress}\right)$, was applied to the Fashion-MNIST dataset. In this case, as with the MNIST dataset, the new classes were confirmed to be available. However, in the green box, in some cases, $c_{\mathrm{new}}$ is no more successful than that in MNIST.

Figure 4 shows the confusion matrices for pairs $\left(c_{f},c_{\mathrm{new}}\right)$ for MNIST and Fashion-MNIST as (8,3) and (Bag, Dress), respectively. The green box in the confusion matrix shows that $c_{\mathrm{new}}$ is properly learned. Meanwhile, the yellow box confirms that appropriate learning has not been performed because no training data have been provided. Therefore, quantitatively, it was confirmed that the more complex the dataset, the more difficult it becomes to learn $c_{\mathrm{new}}$ via the proposed method. These trends are similar when $\left(c_{f},c_{\mathrm{new}}\right)$ are changed in each dataset. To this end, implementing CIL with the replacement of unnecessary classes is realizable.

### 5.2. *Forget and Learn* Method

#### 5.2.1. Forget

Figure 5a shows the experimental results at the point where the existing class is forgotten by replacing $c_{f}$ with the class of noise images based on the setting $\left(c_{f},c_{\mathrm{new}}\right)=\left(8,3\right)$. In the green box in this figure, the output of the existing model for $c_{f}$ corresponds to a noise image, suggesting that $c_{f}$ is not available. In contrast, in the yellow box, no training data are provided for $c_{\mathrm{new}}$ at this point. Figure 5b shows the results of an experiment applying a similar setup as $\left(c_{f},c_{\mathrm{new}}\right)=\left(\mathrm{Bag},\mathrm{Dress}\right)$ for the Fashion-MNIST dataset. This figure confirms the same trend as that for the MNIST dataset, with the output for the existing class $c_{f}$ displaying a noise image.

Furthermore, Figure 6 shows the MNIST and Fashion-MNIST confusion matrices at the point where $\left(c_{f},c_{\mathrm{new}}\right)$ is set to (8,3) and (Bag,Dress), respectively. The green box in the confusion matrix shows that, for the Fashion-MNIST dataset, the classifier misclassifies the noise image to the original class. For the MNIST dataset, the classifier also misclassifies the noise image to a particular class. This confirms that these results are due to the classification characteristics of the classifiers. Therefore, the forgetting class $c_{f}$ is treated as a noise image. In the yellow boxes, the outputs for $c_{\mathrm{new}}$ are not provided accurately because of the lack of training data. Furthermore, this analysis quantitatively shows that replacing certain classes with noise makes existing classes $c_{f}$ unavailable.

#### 5.2.2. Learn

Figure 7a shows the results when a new class is trained by replacing the existing model class with a noise image and fine-tuning afterwards based on the setting $\left(c_{f},c_{\mathrm{new}}\right)=\left(8,3\right)$. The yellow box shows that the model learns $c_{\mathrm{new}}$. This indicates that the models learned a new class in CIL. The green box shows that $c_{f}$ is blurred, although we can see a number-like shape. Figure 7b shows the results on the Fashion-MNIST dataset based on the same setting: $\left(c_{f},c_{\mathrm{new}}\right)=\left(\mathrm{Bag},\mathrm{Dress}\right)$. Similar to the case for the MNIST dataset, the existing $c_{f}$ is displayed as a noise image, and the model learns $c_{\mathrm{new}}$. This indicates that this process is independent of the datasets.

Figure 8 shows the confusion matrix for the fine-tuning process after forgetting. The yellow box shows that the models learn $c_{\mathrm{new}}$ quantitatively. This analysis confirms that the model learns new classes even when $c_{f}$ is replaced by a noise image. However, the performance deteriorates for some data, such as the label “Shirt”.

#### 5.2.3. Performance in Learning New Classes

Figure 9a shows the change in performance for $c_{\mathrm{new}}$ when $c_{f}$ and $c_{\mathrm{new}}$ are changed. The figure has an overall blue color, which indicates an overall improvement in the performance of $c_{\mathrm{new}}$, attributed to the replacement of the existing classes with the class of noise image. Therefore, the effectiveness of the proposed method is confirmed. However, for certain combinations of $\left(c_{f},c_{\mathrm{new}}\right)=\left(7,1\right)$, the performance deteriorates. This improvement suggests that the trend did not change for the Fashion-MNIST dataset. This indicates that the improvement in accuracy achieved by replacing the existing classes with noise is independent of the dataset.

## 6. Discussion

The effectiveness of the proposed method was confirmed through experiments. Specifically, in the *replace* method, experiments on the MNIST and Fashion-MNIST datasets showed that the accuracy of these methods varied across datasets. Conversely, the *forget and learn* method showed variation in the performance of the model in each process. Through these experiments, the advantages and specifics of each method were discussed. However, there are some points to be mentioned regarding the proposed method. This discussion can be summarized in the following points.

**Limited experimental environment:** This study is limited to experiments using VAEs, MNIST and Fashion-MNIST. Training complex and highly accurate generative models is time-consuming. This is the same with the datasets. Learning with more complex datasets than those used is difficult to evaluate comprehensively. Furthermore, the relationships between words and classes in these models are complex. Hence, in this study, to explore the effectiveness of this method for these models, simple and comprehensive experiments were conducted. The results confirm the change in accuracy of the model in the proposed method. However, these results do not highlight the performance on very complex datasets and the models need to be validated in more complex environments in our future research. In addition, as Figure 4b and Figure 6b show, a clear accuracy deterioration in the Pullover, Coat, Sandal, and Dress categories is observed when the proposed methods are applied. This suggests that the performance of the model may degrade due to the natural characteristics of SA, regardless of the image characteristics.

**Lack of general continual learning experiments:** This study presents experimental results by focusing on fine-tuning; however, examples of our applying our method to continual learning methods such as EWC are not provided. Continual learning is an approach for retaining important information regarding existing classes. However, the “stability–plasticity dilemma” arises, in which attempting to retain important information prevents the learning of new classes. Therefore, this method is considered more effective for methods such as continual learning than fine-tuning. Specifically, this method facilitates the learning of new classes by appropriately transforming unnecessary information into noise information. We experimentally confirm that EWC improves the learning accuracy of new classes compared with fine-tuning [50]. Because this method only uses the information in the model, it can be applied to various continual learning techniques. However, it may not be suitable for all continual learning methods; further studies are required on this aspect.

In future works, we aim to expand the experimental environment and apply the proposed method to a wide range of continual learning frameworks with more complex datasets. Additionally, we plan to explore the application of different machine unlearning methods and investigate the impact of various CIL techniques on machine unlearning. This will involve the examination of the balance between forgetting and learning, aiming to understand how these processes facilitate performance improvements in learning new classes. Furthermore, we intend to conduct comparative studies across forgetting techniques to identify their strengths and weaknesses in our future works.

## 7. Conclusions

We proposed SA-based replacement and forgetting methods for CIL and analyzed their impacts on the generative ability. The *replace* method replaces existing classes in the learned model with new classes, enabling the effective learning of new classes. Experiments on the MNIST and Fashion-MNIST datasets revealed that the accuracy of the methods varied with the dataset. On the other hand, the *forget and learn* method increased the effectiveness of the method and led to a change in the performance of the model in each process. The results of this study will provide a foundation for exploiting the internal class structure of models to improve the accuracy of large-scale models and to facilitate continual learning. Furthermore, future work will explore the validation of the proposed approach with more complex class structures and its application to methods with different learning processes and model architectures.

## Figures and Tables

**Figure 1 sensors-24-03087-f001:**
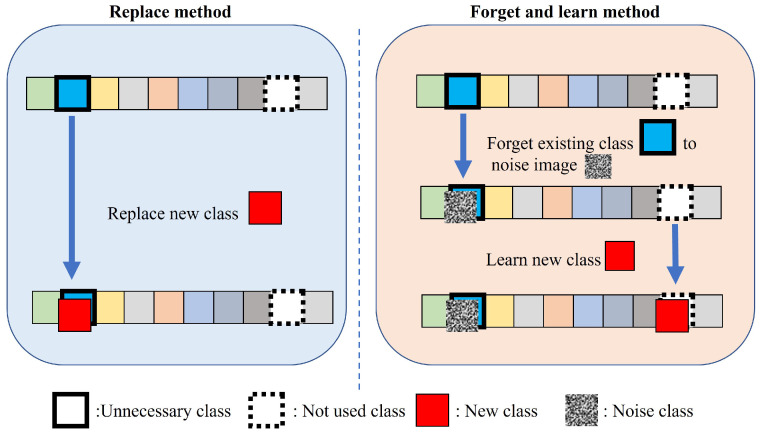
Class transition diagram of the proposed method in class incremental learning.

**Figure 2 sensors-24-03087-f002:**
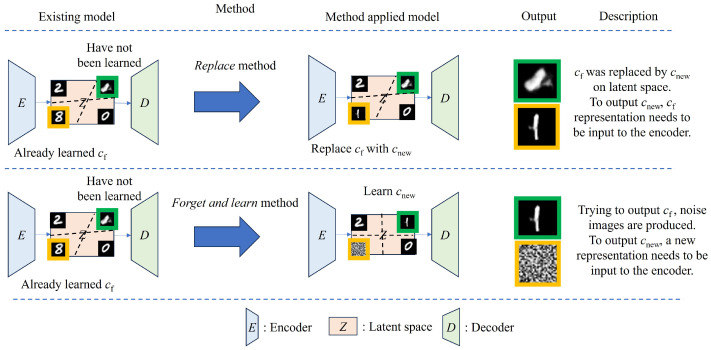
Overview of the proposed method. The model comprises an encoder *E*, a decoder *D*, and a latent space *Z*. The existing model recognizes retained and forgets classes $c_{f}$; however, it does not account for new classes $c_{\mathrm{new}}$. The *replace* method integrates $c_{\mathrm{new}}$ directly into the latent space of $c_{f}$, thereby enhancing the model’s ability to recognize new classes. Conversely, the *forget and learn* method deliberately introduces noise images into the latent space of $c_{r}$ and subsequently trains $c_{\mathrm{new}}$. This method allows the model to partially forget old information while learning new knowledge, facilitating a balanced update.

**Figure 3 sensors-24-03087-f003:**
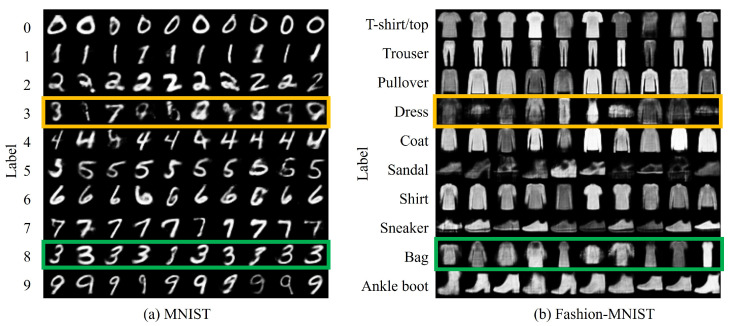
Qualitative evaluation of the proposed *replace* method using the MNIST and Fashion-MNIST datasets, showing outputs for each of the ten classes with ten images per class. The green box indicates the output when unnecessary classes are replaced with new ones. The yellow box denotes classes that are not used in training.

**Figure 4 sensors-24-03087-f004:**
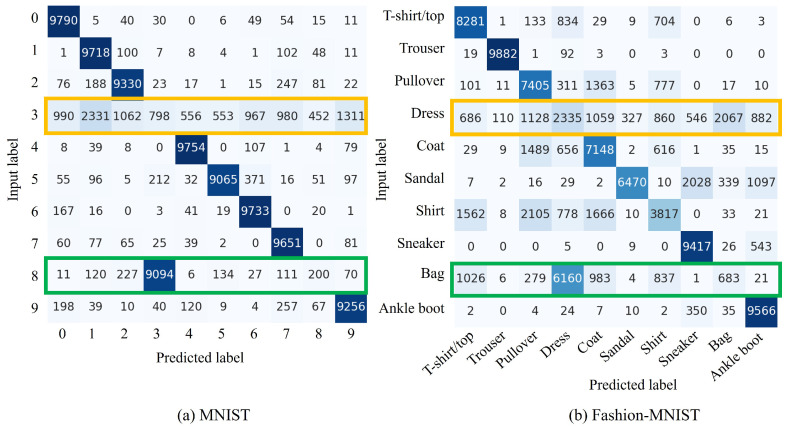
Confusion matrix of the proposed *replace* method using the MNIST and Fashion-MNIST datasets. The vertical axis indicates input labels, while the horizontal axis represents predicted labels. For MNIST, $\left(c_{f},c_{\mathrm{new}}\right)=\left(8,3\right)$ represents the classes to be forgotten and new class, respectively. For Fashion-MNIST, these classes, respectively, correspond to Bag and Dress. The green box highlights the output when an unnecessary class $c_{f}$ is replaced with a new one $c_{\mathrm{new}}$ and the yellow box identifies classes that are not used in this training.

**Figure 5 sensors-24-03087-f005:**
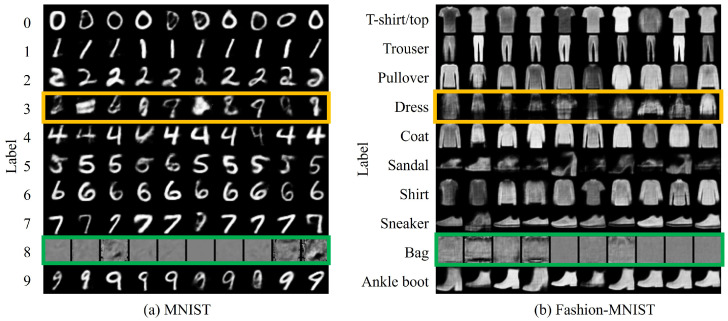
Qualitative evaluation of the proposed *forget and learn* method in terms of introducing a noise image using the MNIST and Fashion-MNIST datasets, showing outputs for each of the 10 classes with 10 images per class. The green box indicates outputs when unnecessary classes are replaced with the class of noise image and the yellow box denotes classes that are not learned at this point.

**Figure 6 sensors-24-03087-f006:**
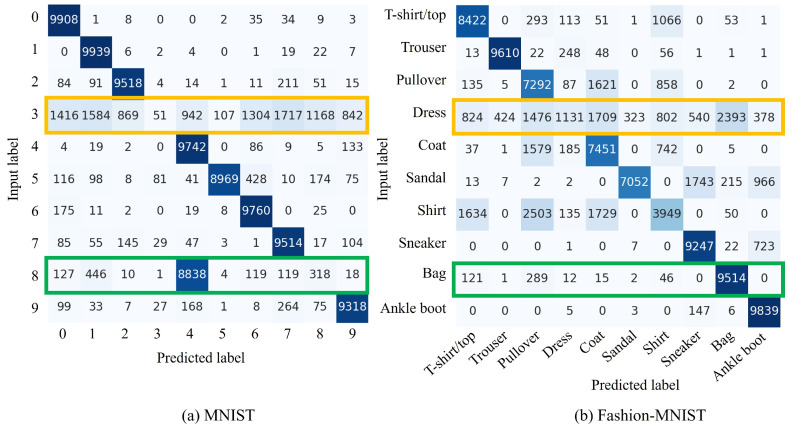
Confusion matrix of the proposed *forget and learn* method for introducing a noise image using the MNIST and Fashion-MNIST datasets. The vertical axis indicates input labels, while the horizontal axis represents predicted labels. For MNIST, $\left(c_{f},c_{\mathrm{new}}\right)=\left(8,3\right)$ represents the classes to be forgotten and new classes, respectively. For Fashion-MNIST, these classes, respectively, correspond to (Bag, Dress). The green box demonstrates the output when $c_{f}$ is replaced with the class of noise images and the yellow box denotes classes that are not used at this point.

**Figure 7 sensors-24-03087-f007:**
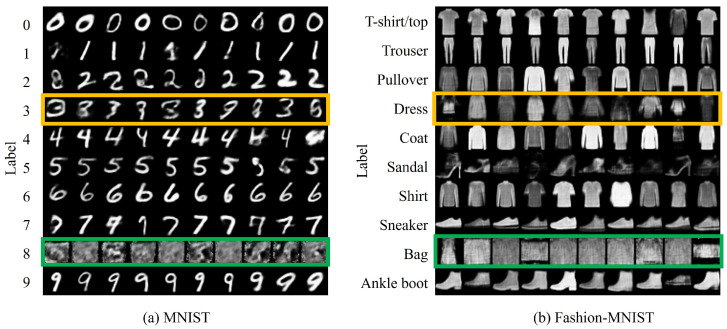
Qualitative evaluation of the proposed *forget and learn* method in terms of its ability to introduce $c_{\mathrm{new}}$ with fine-tuning on the MNIST and Fashion-MNIST datasets, with outputs displayed for 10 classes each with 10 images. The vertical axis indicates the classes. The green box shows the results when $c_{f}$ is replaced with the noise image class, and the yellow box highlights the newly learned class, $c_{\mathrm{new}}$.

**Figure 8 sensors-24-03087-f008:**
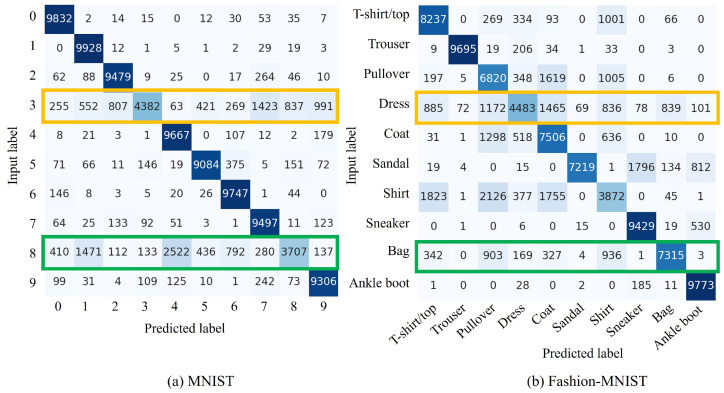
Confusion matrix of the proposed *forget and learn* method for the introduction of $c_{\mathrm{new}}$ with fine-tuning using the MNIST and Fashion-MNIST datasets. The vertical axis indicates input labels, while the horizontal axis represents predicted labels. For MNIST, $\left(c_{f},c_{\mathrm{new}}\right)=\left(8,3\right)$ represents the classes to be forgotten and new classes, respectively. For Fashion-MNIST, these classes correspond to Bag and Dress, respectively. The green box displays outputs when an unnecessary class $c_{f}$ is replaced with the class of noise image and the yellow box denotes the newly learned class $c_{\mathrm{new}}$.

**Figure 9 sensors-24-03087-f009:**
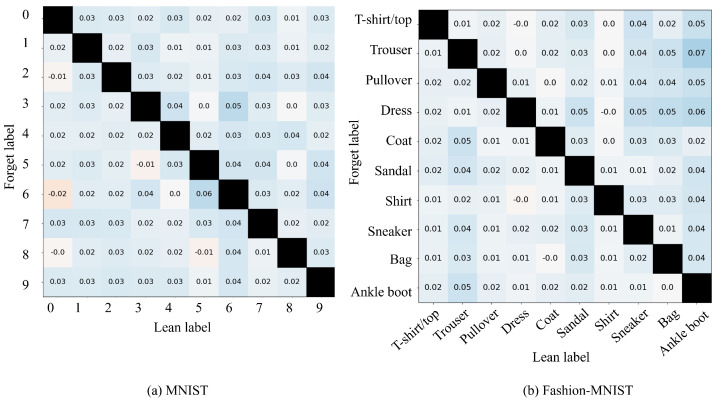
Comparison of the accuracy difference for the newly learned class, $c_{\mathrm{new}}$, with and without the *forget and learn* method. The vertical axis represents $c_{f}$ and the horizontal axis represents the newly learned class, $c_{\mathrm{new}}$. The numbers within the cells indicate the change in accuracy for $c_{\mathrm{new}}$, attributed to the application of the proposed *forget and learn* method. The blue color denotes an improvement in accuracy, red signifies a deterioration, and black indicates no data.

## Data Availability

A publicly available datasets were used in this work. The datasets can be found here: http://yann.lecun.com/exdb/mnist and https://github.com/zalandoresearch/fashion-mnist?tab=readme-ov-fil (accessed on 20 March 2024).

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
