# Peer review of "Analysis of Continual Learning Techniques for Image Generative Models with Learned Class Information Management"

_sensors, 2024, doi:10.3390/s24103087_

Round 1
Reviewer 1 Report
Comments and Suggestions for Authors
Overall, the article appears to be well-written, with reasonable scientific evidence, relatively clear language, and results to support the claims.
There are a couple of passages where you can improve readability or correct language errors, for example: line 35: Approaches that allow the model to process new knowledge while preventing line 67: In a CIL context, replacement methods are very useful.
Figure 1: On the right side of the image, the text says that the existing (with dotted border) class is replaced by a noise class, but in the image in the first step, the noise class is replaced by a "junk class" (solid border). It feels like either the image is wrong or the text description is unclear.
Figure 2: The problem with this image is that it illustrates the words "replace", "forget", etc., but doesn't actually illustrate anything meaningful as the proposed methods named here using those words work.
In Equation 2, “C” was previously entered as the set of all classes rather than as a number. And if it is used differently in Equation 2, then it is not defined.
line 382: in the green box, in some cases cnew is no more successful than in MNIST Caption for Figure 7: The green box shows the results when cf is replaced with the Oise image class, and the yellow box highlights the newly learned class.
One very noticeable issue in this paper is that the methods and techniques described are mainly developed in response to the complexity of (re)training large models with massive training data, yet in this paper the experiments are carried out on relatively small and simple data. datasets. Although this does not disprove the reliability of the presented methodology, it does not give rise to high confidence in it.
A similar problem arises with choosing a model to test, but the authors address it in the discussion section. Another problem is that at the beginning of the article the authors repeat somewhat "intuitive" but relatively unhelpful descriptions of the proposed methods and only later, around line 298, give more precise and formal definitions.
Because these formal descriptions are relatively simple and do not require separate detailed explanation, the reader would benefit if the authors first provided technical definitions and then supplemented them with more intuitive and/or visual aids.
Reviewer 2 Report
Comments and Suggestions for Authors
1. It is recommended to further emphasize the differences and advantages of the proposed method and existing methods to highlight the novelty and readability of the paper.
2. Adopts a different method from SA, focusing on correcting the discriminant space of the training model and its impact on classification in the context of continuous learning. Please describe the discriminant space correction method and forgetting mechanism used in this method in detail to help readers understand.
3. Introduced two methods of modifying deep generative models, "replace" and "forget and learn". Please elaborate further on the relationship and differences between these two methods.
4. Explain how the new class c_new is determined during the process of updating the model by changing the existing class using the replace method. In the forgetting and learning methods, how are the noisy images obtained and what are the specific requirements?
5. How is the balance between forgetting and learning achieved?
6. Qualitatively evaluated the quality and diversity of generated images, and quantitatively evaluated the accuracy of image classification. However, there is a lack of comparative analysis with other advanced models or benchmark models to evaluate the advantages and disadvantages of the model proposed in the manuscript.
7. There are some of the latest research papers on machine learning. It is recommended to read and cite (if applicable) what is helpful to the revision.
--Sotiropoulou, K.F., Vavatsikos, A.P. A Decision-Making Framework for Spatial Multicriteria Suitability Analysis using PROMETHEE II and k Nearest Neighbor Machine Learning Models.
--Bouslihim, Y., Kharrou, M.H., Miftah, A. et al. Comparing Pan-sharpened Landsat-9 and Sentinel-2 for Land-Use Classification Using Machine Learning Classifiers.
Comments on the Quality of English LanguageMinor editing of English language required.
